# Therapeutic Drug Monitoring of Dalbavancin in Real Life: A Two-Year Experience

**DOI:** 10.3390/antibiotics13010020

**Published:** 2023-12-24

**Authors:** Dario Cattaneo, Marta Fusi, Marta Colaneri, Chiara Fusetti, Camilla Genovese, Riccardo Giorgi, Maddalena Matone, Stefania Merli, Francesco Petri, Andrea Gori

**Affiliations:** 1Department of Infectious Diseases, ASST Fatebenefratelli Sacco University Hospital, 20157 Milan, Italy; marta.colaneri@gmail.com (M.C.); chiara.fusetti@unimi.it (C.F.); camilla.genovese@asst-fbf-sacco.it (C.G.); riccardo.giorgi@asst-fbf-sacco.it (R.G.); maddalena.matone@asst-fbf-sacco.it (M.M.); stefania.merli@asst-fbf-sacco.it (S.M.); francesco.petri@unimi.it (F.P.); andrea.gori@unimi.it (A.G.); 2Department of Biomedical and Clinical Sciences, Università degli Studi di Milano, 20157 Milan, Italy; marta.fusi@unimi.it

**Keywords:** dalbavancin, therapeutic drug monitoring, osteoarticular infections, periprosthetic joint infections

## Abstract

Dalbavancin is a long-acting lipoglycopeptide that is registered for the treatment of acute bacterial skin and skin structure infections, and it is also increasingly used for infections that require prolonged antibiotic treatment. Here, we present the results from the first 2 years of a service set up in December 2021 for the therapeutic drug monitoring (TDM) of dalbavancin in clinical settings. In particular, we compared the trough concentration (Cmin) to maximum concentration (Cmax) in patients with osteoarticular infections receiving prolonged treatment with dalbavancin. Log-linear regression models were used to estimate the timing of dalbavancin administration with the goal of maintaining Cmin concentrations of >8 mg/L in the two TDM-based strategies. From December 2021 to November 2023, 366 TDMs of dalbavancin from 81 patients were performed. The Cmin and Cmax concentrations of dalbavancin ranged from 4.1 to 70.5 mg/L and from 74.9 to 995.6 mg/L, respectively. With log-linear regression models, we estimated that each injection should be administered every 42–48 days to maintain the Cmin concentrations. Out of the 81 patients, 37 received at least three doses of dalbavancin for the treatment of osteoarticular infections. Despite there being no significant differences in the days of dalbavancin treatment (130 ± 97 versus 106 ± 102 days), the patients in the Cmax-based TDM group received a significantly lower number of dalbavancin injections (5.2 ± 1.8 versus 7.3 ± 2.6 injections, *p* = 0.005), and they were administered over a longer period of time (40 ± 10 versus 29 ± 14 days, *p* = 0.013) than in the Cmin-based TDM group. In conclusion, Cmax-based TDM was associated with a significant reduction in the inter-individual variability of dalbavancin concentrations and lower drug dosing frequency than those of Cmin-based TDM. This approach could, therefore, favor a more rational and targeted use of dalbavancin in patients requiring prolonged treatment.

## 1. Introduction

Dalbavancin is a novel lipoglycopeptide characterized by a terminal half-life of >14 days, allowing the administration of either a two-dose regimen (1000 mg day 1, 500 mg day 8) or a single-dose regimen of 1500 mg for the treatment of acute bacterial skin and skin structure infections (ABSSSIs) [1,2,3,4]. After its initial distribution, its extensive diffusion into body tissues, including bone and articular tissues, causes the drug levels in plasma to rapidly decline over the first 48 h with a total volume of distribution of nearly 16 L [2,3]. Accordingly, dalbavancin (off-label) is also increasingly used for the treatment of patients with osteoarticular infections or periprosthetic joint infections [1,5,6,7]. Moreover, dalbavancin has been shown to have important activities in in vitro models of experimental endocarditis due to *Staphylococcus aureus* with or without reduced susceptibility to vancomycin and teicoplanin [8]. This evidence provided the rationale for clinical studies that documented the efficacy and cost-effectiveness of dalbavancin for the sequential treatment of patients with infective endocarditis [9,10,11].

Through Monte Carlo simulations, Cojutti et al. recently determined that a biweekly dosing regimen of 1500 mg of dalbavancin may sustain its efficacy for up to 5 weeks in patients with staphylococcal osteoarticular infections [12]. Pillar studies by Pea et al. subsequently provided preliminary evidence that maintaining dalbavancin trough concentrations (Cmin) of ≥4 or ≥8 mg/L over time could represent efficacy thresholds in PK/PD [13,14,15]. These proof-of-concept studies demonstrated the usefulness of Cmin-based therapeutic drug monitoring (TDM) as a tool for optimizing dalbavancin dosing in prolonged antibiotic treatments for complicated osteoarticular infections. As an alternative approach, other authors proposed the maximum dalbavancin concentration (Cmax) as the optimal single sampling point for estimating the drug’s area under the curve and as the ideal dalbavancin posology for patients requiring prolonged antibiotic therapy [16,17].

An expert review panel recently published a consensus document on the dose regimen and TDM for the long-term use of dalbavancin. The goal was to collect proposals that accommodated different healthcare settings and levels of resource availability, and it was centered around the duration of dalbavancin treatment [18]. They suggested the achievement of adequate dalbavancin concentrations for up to 6 weeks; 3000 mg of dalbavancin should be given over 4 weeks for complex infections requiring >2 weeks of treatment. The experts also pointed out that the TDM of dalbavacin is advised for longer treatment durations and in cases of renal failure.

In December 2021, we established TDM for dalbavancin concentrations as a diagnostic service for patients who were referred to our laboratory, and we also included samples sent by external hospitals. Based on their practices, physicians used TDM based on the measurement of the Cmin and/or Cmax of dalbavancin. Here, we present the results of the first 2 years of the TDM of dalbavancin.

## 2. Results

### 2.1. Patients’ Characteristics

From December 2021 to November 2023, 366 instances of TDM for dalbavancin were performed in our laboratory for 81 patients. The patients were mostly males (62%), with a mean age of 63 ± 19 years, and they received an average of 4.8 ± 2.7 dalbavancin injections (mean dose: 1436 ± 201 mg/injection). The mean serum creatinine, estimated glomerular filtration rate, serum albumin, and serum aspartate aminotransferase were, respectively, 1.3 ± 1.3 mg/dL, 78 ± 29 mL/min, 35 ± 5 g/L, and 26 ± 15 U/L. Patients were administered dalbavancin at 1500 mg (i.e., day 1 and day 7; eventually repeated if needed based on the TDM results). Patients with GFR < 30 mL/min were treated with 1000 mg (day 1), followed by maintenance doses of 500 mg.

### 2.2. Overall Distribution of Dalbavancin Concentrations

The Cmin for dalbavancin ranged from 4.1 to 70.5 mg/L (mean 20.7 ± 14.0 mg/L). Figure 1 illustrates the highest dalbavancin concentrations (39.6 ± 35.4 mg/L), and the largest pharmacokinetic inter-individual variability (CV%: 89%) was observed before the second injection, which was performed at a mean of 8 ± 3 days after the first injection. A highly significant reduction (*p* < 0.001) in the Cmin for dalbavancin and in the corresponding pharmacokinetic variability was observed in the following injections: 60%, 50%, 57%, 42%, and 51% from the third to more than six injections (Figure 1).

One-hundred and seven of the TDM assessments indicated the Cmax of dalbavancin, which ranged from 74.9 to 995.6 mg/L (mean: 309.8 ± 158.4 mg/L). Coupled with the Cmin values, these samples facilitated the development of log-linear regression models describing the time distribution of dalbavancin concentrations for each injection. By targeting the Cmin values for dalbavancin that were not lower than 4 or 8 mg/L, we were able to estimate the number of days needed between each injection to guarantee these targets. Table 1 demonstrates that each injection should be performed every 42–48 days to maintain the Cmin for dalbavancin at >8 mg/L, with important individual differences, as exemplified by Figure 2.

Two patients who were treated with the same dalbavancin dose (1500 mg/injection) for a comparable treatment time of approximately 8 months exhibited distinct injection schedules. However, by estimating the time for the next drug injection using the models described above, we administered seven dalbavancin doses every 36 ± 5 days for patient A and five dalbavacin doses every 61 ± 14 days for patient B.

### 2.3. TDM of Dalbavancin for Osteoarticular Infections

Thirty-seven out of the 81 patients (35% women, mean age: 67 ± 14 years) received at least three doses of dalbavancin for the treatment of osteoarticular and/or periprosthetic joint infections caused mainly by methicillin-resistant *Staphylococcus Aureus* (50%) and methicillin-sensitive *Staphylococcus Aureus* (35%). The patients in the Cmax-based TDM group (n = 17) were mainly males (78%) and were significantly older (73 ± 12 vs. 67 ± 14; *p* = 0.006) than those in the Cmin-based TDM group (n = 18, 53% males). No significant differences were found in the mean dalbavancin doses between the two groups; 17% and 19% of the patients in the Cmin- and Cmax-based TDM groups, respectively, were given 1000 mg (day 1) followed by 500 mg of dalbavancin, whereas all of the remaining patients were given 1500 mg of dalbavancin per injection. The measured dalbavancin Cmin values ranged from 5.3 to 56.0 and from 5.4 to 57.3 mg/L, respectively, in the Cmin- and Cmax-based TDM groups, with no differences in the mean drug Cmin values (19 ± 10 vs. 16 ± 11 mg/L; *p* = 0.116). No dalbavancin Cmin values were <4 mg/L; 8.7% of the TDM results were found to be <8 mg/L, with no differences between groups (Table 2).

The enrolled patients received a mean of 6.5 ± 2.5 injections of dalbavancin for osteoarticular infections. Despite there being no significant differences in the overall number of days of dalbavancin treatment (130 ± 97 versus 106 ± 102 days, *p* = 0.183) and the maximum treatment duration (419 days for both groups), the patients in the Cmax-based TDM group received significantly fewer dalbavancin injections (5.2 ± 1.8 versus 7.3 ± 2.6 injections, *p* < 0.0001), and they were administered over an extended period (40 ± 10 versus 29 ± 14 days, *p* = 0.013) compared with the Cmin-based TDM group (Table 2). In additional analyses, we compared the data on the pharmacokinetics of dalbavancin between the Cmin- and Cmax-based TDM groups. The dalbavancin Cmin values were consistently lower in the Cmax-based TDM group than in the Cmin-based TDM group in each comparison, reaching statistical significance on some occasions (Table 3). The adoption of Cmax-based TDM resulted in a significant increase in the time interval between two injections compared to that in Cmin-based TDM (ranging from 10 to 15 days depending on the injection). As shown in Table 3 (and visualized graphically in Figure 3), Cmax-based TDM was also associated with a trend of reduced inter-individual variability in dalbavancin Cmin values compared with that in Cmin-based TDM.

## 3. Materials and Methods

### 3.1. Study Population and Study Design

This study involved a retrospective analysis of TDM that was routinely carried out by the ASST Fatebenefratelli Sacco University Hospital (Milan, Italy) from December 2021 to November 2023; the analysis included samples that were collected within our hospital and those received from other hospitals (our laboratory is a reference laboratory for the TDM of dalbavancin in Lombardy Region, Italy). This service also included the on-demand estimation of the next dalbavancin injection. This estimation required the collection of both the Cmin and the Cmax for dalbavancin. It was not uniformly used by physicians, and two cohorts of patients were generated: one cohort in which the TDM was based on measurements of both the Cmin and the Cmax (with the estimation of the next dalbavancin dose being performed by our service) and one cohort in which the TDM was based on the Cmin measurement (in this case, we did not provide estimations, and physicians decided the time for the next injection).

The first part of this study outlined the comprehensive data from the TDM of dalbavancin performed in our laboratory; in the second part of the study, we focused on adult patients with osteoarticular infections receiving at least 3 doses of dalbavancin while comparing Cmin-based and Cmax-based TDM in terms of days between injections and the percentage of samples below therapeutic thresholds, as defined in the available literature (more details are given in the next section). 

This retrospective study utilized anonymized data that were collected for clinical purposes in accordance with the requirements of the Italian Personal Data Protection Code. Approval from an ethics committee was considered unnecessary as, per the Italian law, approval is only required in the case of prospective clinical trials of medical products for clinical use (Articles 6 and 9 of Legislative Decree No. 211/2003). Written informed patient consent for medical procedures/interventions performed according to clinical practice was collected by each center.

### 3.2. TDM of Dalbavancin in Patients with Osteoarticular Infections

We retrospectively analyzed data from adult patients who received at least 3 doses of dalbavancin for the treatment of osteoarticular and/or periprosthetic joint infections for whom TDM was performed as part of clinical practice. On the day of the scheduled injection, the Cmin for dalbavancin within a 1 h time window before the injection was considered for both Cmin- and Cmax-based TDM. Peak drug concentrations (30 to 60 min after the end of the infusion) were collected only in patients subjected to Cmax-based TDM. Cmin values were used to assess the timing accuracy of the previous administration (by assessing the number of samples with concentrations below 4 or 8 mg/L), whereas the Cmax for dalbavancin was used in log-linear regression models to estimate the timing of the next injection with the aim of maintaining the Cmin for dalbavancin at, respectively, ≥4 or >8 mg/L, as suggested by the recent literature [12,13,14,15,16,17,18].

### 3.3. Assessment of Plasma Dalbavancin Concentrations

Plasma blood samples with dalbavancin were assessed by using ethylene diamine tetra-acetic acid (EDTA)-containing Vacutainers^®^. All samples were centrifuged at 3000× *g,* and the plasma was separated and stored at −20 °C. Dalbavancin quantification was performed through a validated LC-MS/MS method that was developed and validated according to the EMA guidelines. The analytical process consisted of a fast protein precipitation protocol of 50 µL of plasma with 400 µL of precipitation solution (methanol/acetonitrile 3/1), centrifugation at 10,000× *g*, 1:3 dilutions with water, and analysis. Chromatographic separation was achieved using a gradient (acetonitrile and water with formic acid 0.1%) on a reversed-phase analytical column (Acquity UPLC BEH C18 1.70 μm 2.1 × 50 mm; Waters, Milan, Italy). For quantification, analysis was performed in the ESI-positive mode by monitoring the transition (*m*/*z* = 909.45 > 340.2) for dalbavancin. The method was linear from 1 to 500 mg/L, with intraday and interday assay imprecision and inaccuracy consistently being <10% during each analytical run.

### 3.4. Statistical Analyses

Dalbavacin concentrations were considered both as a continuous variable (expressed as the mean ± standard deviation) and categorical variables, and data (expressed as a percentage) were clustered according to their frequency distribution within the therapeutic ranges proposed in the available literature (>4 or >8 mg/L). Inter-individual variability in the dalbavancin concentrations was calculated as the percentage of the coefficient of variation (CV%). Log-linear regressions were performed using the built-in statistical package in Microsoft Excel (Microsoft Office Software, Microsoft Company, Redmond, WA, USA). The goodness of the log-linear regression models developed by plotting logarithmic concentrations (Y) of dalbavancin with time (X) was quantified using the coefficient of correlation (r). Unpaired *t*-tests were used to compare the Cmin for dalbavancin, the number of drug injections, and the times of injections between Cmin- and Cmax-based TDM. Statistical significance was set at <0.05 (significant difference) and <0.01 (highly significant difference).

## 4. Discussion

When used at the approved one- or two-dose regimens for the treatment of ABSSSIs, dalbavancin typically does not benefit from TDM for its plasma concentrations due to the short length of the treatment. However, growing evidence showing the potential role of TDM in other clinical settings is now available. In 2020, the application of TDM in three critically ill patients treated with dalbavancin for severe necrotizing fasciitis revealed its capacity to identify rapid drug clearance associated with suboptimal dalbavancin exposure [19]. Using this approach, we were able to identify a severely hypoalbuminemic patient with such an association. Subsequently, various authors have reinforced the relevance of TDM in estimating the duration of optimal target attainment for dalbavancin in staphylococcal osteoarticular infections [12,13,20]. Notably, Stroffolini et al. reported their experience with the TDM of dalbavancin in treatment-experienced patients with skin, osteoarticular, or vascular infections [17]. Remarkably, they reported significant differences in the dalbavancin concentrations between male and female patients. More recently, Hervochon et al. reported the TDM data of 133 patients from 13 French hospitals treated with 1500 mg doses and followed for up to 6 weeks after the last dalbavancin doses [21]. They found that dalbavancin concentrations were significantly affected by the renal function (which was significantly higher in patients with GFR < 60 mL/min) and by the body weight (which was significantly lower in patients weighing >75 kg).

Collectively, these findings highlight the growing interest in TDM for dalbavancin in day-by-day clinical practice. Our two-year experience with the TDM of dalbavancin aligns with this trend. Indeed, we observed a progressive increment in TDM requests for dalbavancin, which were mainly for patients that required more than two drug injections in our hospital. The large and predominant treatment regimen involved two doses of 1500 mg of dalbavancin at a distance of around 8 days. The Cmin levels measured before the second injection were associated with higher values and the largest inter-individual pharmacokinetic variability (close to 90%). Nevertheless, the Cmin dalbavancin concentrations measured before the second injection, in conjunction with the Cmax measured after the second injection (when available), were used to build predictive models that were capable of estimating the time of the third injection (if needed) by using target Cmin dalbavancin concentrations above 4 or 8 mg/L as a target, as suggested in the current literature [12,13,14,15,16,17,18]. The same approach was used to estimate the time lag for the following injections, at least for patients who required a prolonged antibiotic treatment. Overall, dalbavancin injections should be administered at a mean of every 42–48 days to maintain concentrations of >8 mg/L, depending on the injection considered. While this information could be potentially useful as a starting point for centers that lack the possibility to perform TDM of dalbavancin, it is important to underline that the data derived from log-linear regression models that were obtained were pooled from all patients, and this can provide information on the population pharmacokinetics, which have significant individual variation. This concept is well exemplified in Figure 2. The application of Cmax-based TDM while using individual log-linear regressions resulted in the estimation of greatly different time lags for dalbavancin injections between the two patients (i.e., 36 versus 61 days). Remarkably, this approach also resulted in important differences in the number of dalbavancin injections required to cover the same treatment period (seven versus five injections to cover 8 months of treatment) between the two patients.

Our study indirectly supports the increasing use of dalbavancin in long-term treatments of osteoarticular and/or periprosthetic joint infections, taking advantage of its optimal penetration into bone and joint tissues and its long half-life [2,3]. A recent systematic review affirmed the efficacy of dalbavancin for osteoarticular infections, particularly in patients receiving at least three drug doses [5]. Through Monte Carlo simulations, Cojutti et al. proposed that a dosing regimen of two 1500 mg of dalbavancin one week apart potentially ensured efficacy for up to 5 weeks against staphylococcal osteoarticular infections. A clinical reassessment at that time could inform decisions regarding the need for an additional dose to prolong the effective treatment [12]. The optimal timing of dalbavancin administration after the third dalbavancin injection remains ill-defined.

Our study provides preliminary evidence supporting the feasibility of addressing the individualization of long-term drug-dosing regimens by maintaining optimal dalbavancin exposure. After implementing TDM for dalbavancin in our hospital, we found different patterns of use among physicians based on their clinical practices. Some practitioners assessed the Cmin of dalbavancin on the scheduled injection day to know a posteriori if the planned administration time resulted in adequate dalbavancin exposure. Conversely, other physicians collected an additional sample 30 min after the end of each injection (corresponding to Cmax). This was used to estimate the timing of the next injection through log-linear regression models aimed at maintaining dalbavancin Cmin values of ≥8 mg/L [12,13,14,15,18]. The accuracy of the predictions was verified by measuring the Cmin values at the following injection. During these 2 years of TDM for dalbavancin when treating osteoarticular infections, we observed an almost equal distribution between the two approaches, with a tendency toward more frequent use of Cmax-based TDM than Cmin-based TDM (57% vs. 43%). A potential disadvantage is that this approach requires the collection of a blood sample for the Cmax measurement in addition to the traditional Cmin. However, it should be considered that the simultaneous collection of Cmax and Cmin samples is common practice in TDM for other antibiotics, such as vancomycin and aminoglycosides [22].

For each subsequent injection, the dalbavancin Cmin values were significantly lower in the Cmax-based TDM group than in the Cmin-based group. However, we did not observe Cmin values below 4 mg/L, with nearly 10% of samples from both groups having dalbavancin concentrations below 8 mg/L. This suggested that both Cmin-based and Cmax-based TDM had a comparably low risk of drug underexposure. Furthermore, as shown in a direct comparison of the two TDM strategies during each injection, the Cmax-based approach was associated with a trend toward reduced inter-individual pharmacokinetic variability compared with Cmin-based TDM. Taken together, these findings suggest that both approaches provided optimal exposure to dalbavancin, but Cmax-based TDM might offer a more tailored dalbavancin dosing approach than that of Cmin-based TDM.

Remarkably, most of the patients in the Cmax-based TDM group were treated with dalbavancin injections every 35–45 days, with some experiencing delays of up to 68 days before the next dose. However, nearly all patients in the Cmin-based TDM received scheduled injections every 20–30 days. After a year of treatment, this resulted in a reduction of 1–2 injections per patient in the Cmax-based TDM group. Although a formal cost-effectiveness analysis was not conducted, we are confident that Cmax-based TDM-based optimization of dalbavancin dosing could help control costs for national health services while ensuring sufficient systemic drug exposure.

The potential limitations of our study are represented by the retrospective design, the small sample size, which limited the generalizability of the findings, and the lack of data on clinical outcomes. These preclude the possibility of identifying factors that could significantly impact the variability in dalbavancin concentrations. Assessing clinical outcomes requires longer follow-up and may be affected by several variables, but this is beyond the scope of the present investigation.

## 5. Conclusions

Cmax-based TDM is associated with a reduction in inter-individual variability for dalbavancin concentrations and a lower dosing frequency than that of traditional Cmin-based TDM. This approach facilitates a precise, targeted, and cost-effective use of dalbavancin for prolonged treatments of osteoarticular infections, and the utility of TDM for optimizing therapeutic interventions and controlling the costs associated with the therapy is emphasized.

## Figures and Tables

**Figure 1 antibiotics-13-00020-f001:**
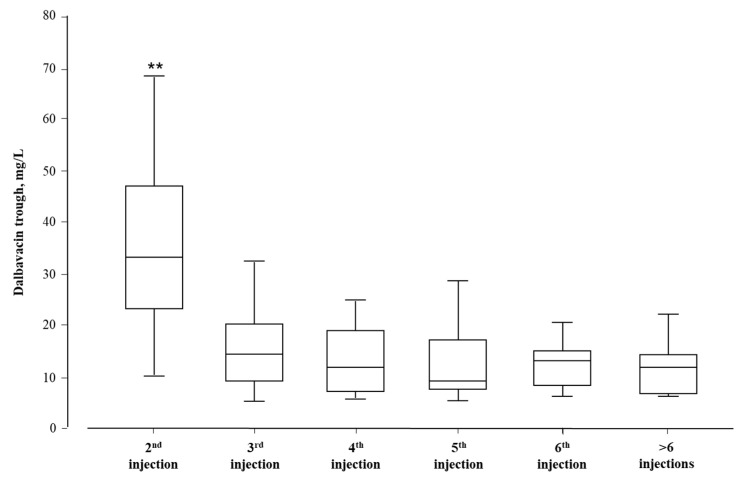
Box plot of the Cmin values for dalbavancin clustered according to the drug injection. ** *p* < 0.001 versus other injections.

**Figure 2 antibiotics-13-00020-f002:**
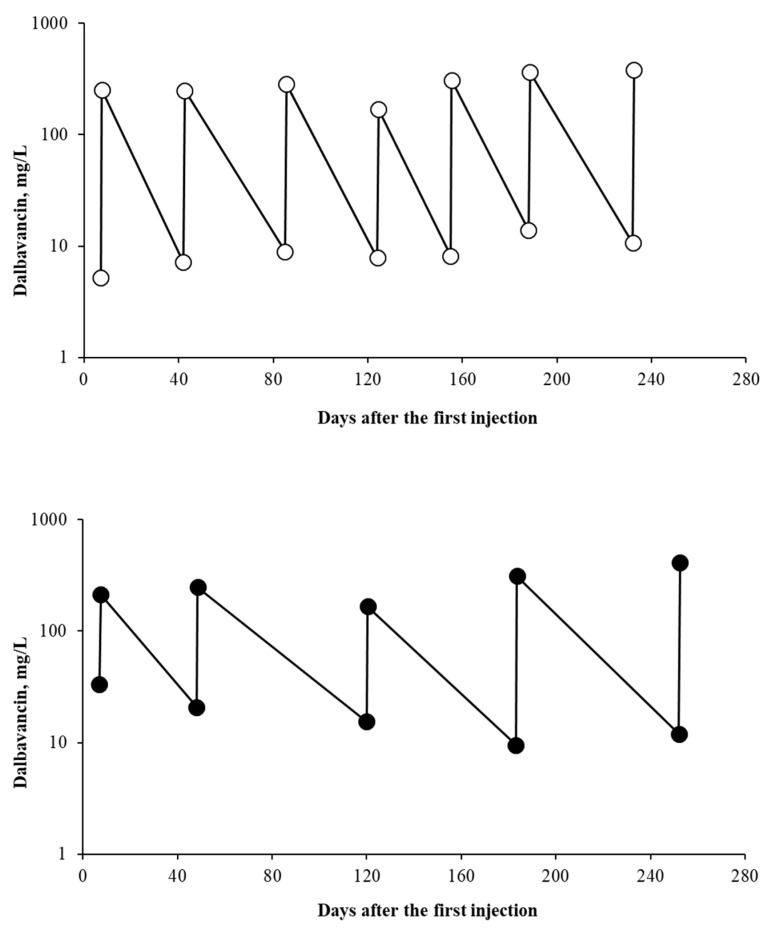
Time course of dalbavancin concentrations and the timing of injections of two patients who underwent Cmax-based therapeutic drug monitoring.

**Figure 3 antibiotics-13-00020-f003:**
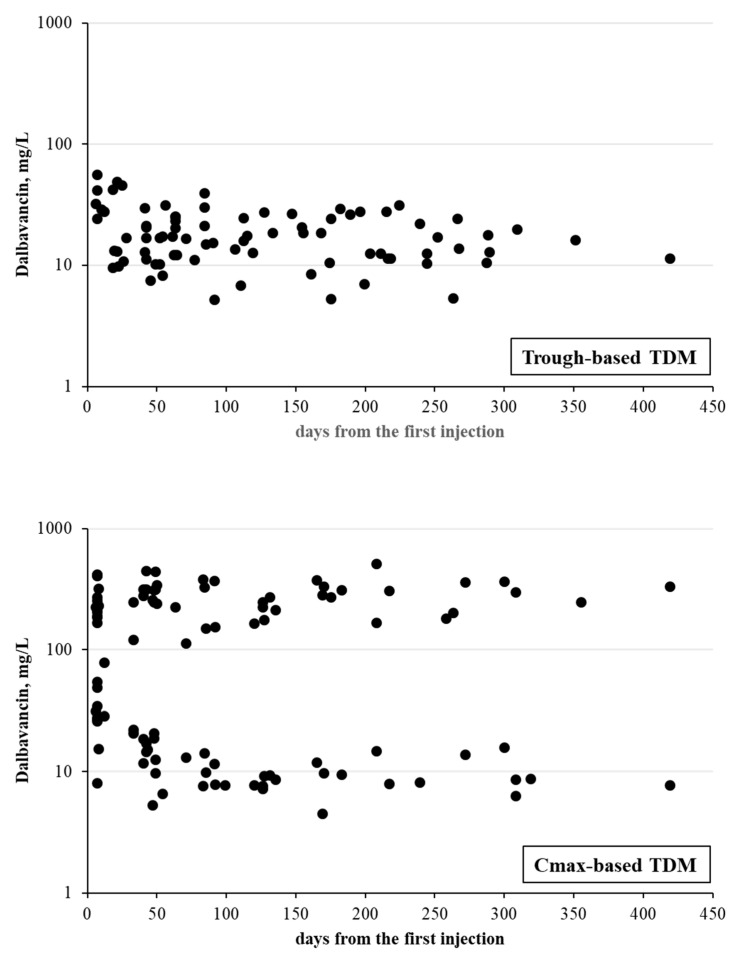
Dalbavancin concentrations in patients with osteoarticular infections (n = 37) who underwent Cmin-based (upper panel) or Cmax-based (lower panel) therapeutic drug monitoring for dalbavancin concentrations.

**Table 1 antibiotics-13-00020-t001:** Log-linear regression models describing the time distribution of dalbavancin concentrations.

Dalbavancin Injection	Log-Linear Regression Model	Correlation Coefficient (r)	Days >4 mg/L	Days >8 mg/L
2nd injection	Y = 189.8 × e^−0.074^	0.878	52	42
3rd injection	Y = 221.6 × e^−0.077^	0.908	52	43
4th injection	Y = 136.7 × e^−0.055^	0.823	59	48
5th injection	Y = 130.3 × e^−0.062^	0.807	56	45
6th injection	Y = 194.5 × e^−0.073^	0.846	53	44
>6 injections	Y = 167.6 × e^−0.072^	0.837	52	42
Overall	Y = 177.2 × e^−0.070^	0.878	54	44

**Table 2 antibiotics-13-00020-t002:** Demographic and pharmacological data of patients undergoing Cmin- or Cmax-based therapeutic monitoring of dalbavancin.

Characteristics	Overall	Cmin-Based TDM	Cmax-Based TDM
Patients, n	37	19	18
Females, %	35%	47%	22% *
Age, years	67 ± 14	64 ± 13	73 ± 12 **
Days of dalbavancin therapy	120 ± 100	130 ± 97	106 ± 102
Maximum drug treatment, days	419	419	419
Dalbavancin dose, mg	1436 ± 201	1460 ± 198	1419 ± 185
Dalbavancin injections, n	6.5 ± 2.5	7.3 ± 2.6	5.2 ± 1.8 **
Days between injections	34 ± 14	29 ± 14	40 ± 10 *
Samples < 4 mg/L, %	0	0	0
Samples < 8 mg/L, %	8.7%	9.3%	7.7%

** *p* < 0.01 and * *p* < 0.05 versus Cmin-based TDM.

**Table 3 antibiotics-13-00020-t003:** Variability in dalbavancin Cmin values in patients undergoing Cmin- or Cmax-based therapeutic monitoring (TDM).

	Cmin-Based TDM		
	Dalbavancin Cmin, mg/L (CV%)	Days from the 1st Injection	Days from the Last Injection (range)
2nd injection	32 ± 14 (45%)	10 ± 6	10 ± 6 (6–12)
3rd injection	14 ± 8 (57%)	37 ± 14	33 ± 20 (14–50)
4th injection	15 ± 7 (43%)	72 ± 30	29 ± 11 (20–35)
5th injection	17 ± 10 (56%)	93 ± 27	32 ± 20 (14–57)
6th injection	15 ± 6 (41%)	120 ± 30	28 ± 33 (21–42)
≥6 injections	19 ± 7 (37%)	236 ± 63	31 ± 15 (14–43)
	**Cmax-based TDM**		
	Dalbavancin Cmin, mg/L (CV%)	Days from the 1st injection	Days from the last injection (range)
2nd injection	30 ± 13 (44%)	8 ± 2	8 ± 2 (6–12)
3rd injection	16 ± 7 (46%)	43 ± 6	36 ± 7 (20–47)
4th injection	11 ± 4 (40%)	100 ± 46 *	41± 3 ** (35–44)
5th injection	8 ± 2 * (21%)	135 ± 18 **	45 ± 13 * (38–72)
6th injection	10 ± 3 (32%)	178 ± 9 *	45 ± 9 ** (39–63)
≥6 injections	11 ± 4 * (32%)	297 ± 63 *	43 ± 10 * (31–64)

** *p* < 0.01 and * *p* < 0.05 versus Cmin-based TDM; CV%: Percent of coefficient of variation.

## Data Availability

The data are contained within the article.

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
