# Peer review of "Therapeutic Drug Monitoring of Dalbavancin in Real Life: A Two-Year Experience"

_antibiotics, 2023, doi:10.3390/antibiotics13010020_

Round 1

Reviewer 1 Report

Comments and Suggestions for Authors

Manuscript antibiotics-2752326-peer-review-v1 describes the application of therapeutic drug monitoring approaches for the dosing of dalbavancin in patients with osteoarticular infections.

The manuscript is well-written and within the scope of Antibiotics. This manuscript revealed a clinically-important findings that that could be used as a guide for the long-term dosing of dalbavancin.  However, there are some comments that need to be addressed before publication:

Major concerns:

1.       The current study was designed as a retrospective study. Yet, the authors have described that they actively intervened the dosing of dalbavancin in two patients (A and B) based on data from the Log-linear regression models. This intervention contradicts the design of the current study.

Minor concerns

1.       Abstract, line 13: “clinical settings. of this. In particular” should read “clinical settings. In particular”.

2.       Table 2, Plasma Dalbavancin C0, mg/L: the values described in row of C0 seem to be very low. The term of plasma C0 is usually used to describe initial plasma concertation which cant be in the range of 17 mg/L (as described in the table) considering the values of Cmax presented in Figure 3.

Author Response

Responses to Reviewer 1

The manuscript is well-written and within the scope of Antibiotics. This manuscript revealed a clinically-important finding that that could be used as a guide for the long-term dosing of dalbavancin.  However, there are some comments that need to be addressed before publication

  1. The current study was designed as a retrospective study. Yet, the authors have described that they actively intervened the dosing of dalbavancin in two patients (A and B) based on data from the Log-linear regression models. This intervention contradicts the design of the current study.

We apologize for the misunderstanding. As better clarified in the revised version of the manuscript, the present study is a retrospective analysis of a 2-year experience with the TDM of dalbavancin. This service, which is routinely carried out in our hospital, includes also the on demand estimation of the time for the next dalbavancin injection. For this estimation we require that physicians collect both the Cmin and the Cmax dalbavancin concentrations. Not all the physicians use this estimation. This is why we described two cohort of patients: one cohort in which TDM was based on both the Cmin and the Cmax measurements (with the estimation of the next dalbavancin dose performed by our service) and one cohort in which TDM was based on Cmin measurement (in this case we do not provide estimations, and physicians decide the time for the next injection).

  1. Abstract, line 13: “clinical settings. of this. In particular” should read “clinical settings. In particular”.

We apologize for the poor English language. The revised manuscript has been revised by a native English speaker.

  1. Table 2, Plasma Dalbavancin C0, mg/L: the values described in row of C0 seem to be very low. The term of plasma C0 is usually used to describe initial plasma concertation which can’t be in the range of 17 mg/L (as described in the table) considering the values of Cmax presented in Figure 3.

The C0 dalbavancin concentrations presented in Table 2 are the mean values measured in all visits. We agree with the Reviewer that this may be misleading from the readers. Accordingly, in the revised Table 2 we have removed the line describing the C0 dalbavancin concentrations.

Reviewer 2 Report

Comments and Suggestions for Authors

Dear,

authors compared trough vs. Cmax base for TDM in patients with osteoarticular infections receiving prolonged treatment with dalbavacin. This is an interesting topic and expect recognized by the wider audience. In addition, some text corrections should be made (ex. abstract line 12, etc). Also, maybe some analysis to identify some variability factors should be performed within both arms. Also, it would be nice to see minimum and maximum values rather than only mean and SD in Table 2 and 3. It is important when the sample size is small.

Sincerely,

Comments on the Quality of English Language

Close reading of text should be done.

Author Response

Responses to Reviewer 2

  1. The authors compared trough vs. Cmax base for TDM in patients with osteoarticular infections receiving prolonged treatment with dalbavacin. This is an interesting topic and expect recognized by the wider audience. In addition, some text corrections should be made (ex. abstract line 12, etc). Also, maybe some analysis to identify some variability factors should be performed within both arms. Also, it would be nice to see minimum and maximum values rather than only mean and SD in Table 2 and 3. It is important when the sample size is small.

We apologize for the poor English language. The revised manuscript has been revised by a native English speaker. Minimum and maximum values of the days from the last injections have been added in the revised Table 3. The limited sample size, which precludes the possibility to perform analysis to identify factors which could significantly impact on the variability in dalbavancin concentrations have been added as a potential study limitation.

Reviewer 3 Report

Comments and Suggestions for Authors

The authors have formulated a well written and important manuscript regarding dalbavancin TDM in clinical practice.

An aim is defined in this retrospective analysis, however some points could use some revision:

1.     Cmax-based and trough based: Why do you chose to mix the terminogly and not standardize to cmax and cmin oder peak an trough?

2.     What software did you use for regression models, statistics and figures? Please state in the method section.

3.     Please explain why you didn’t compare different models for the analysis (Table 1) like it is described in Bhamidipati or why did you chose loglinear since there are different options? Why didn’t you use mean absolute error (MAE)/root mean square error (RMSE) for evaluation?

4.     You describe a retrospective analysis of routine data. So how did you estimate the dosing interval with cmax based TDM within these two years after cmax measurement? What was the standard approach bevor you analysed the log-linear regression models with your data as a result after the 2-year experience? Please describe the routine approach in the method section.

5.     Please check the formatting for the pathogens e.g. line 43. Shouldn’t that be written in italics?

7.     The manufacturer recommends a dose adjustment according to GFR. Please add the GFR for the patients in the demografics.

8.     What was the standard dosing procedure in your hospital? The actual description is quite confusing. 1000 mg and 500 mg or 1500 mg? 

9.     Which pathogens where found in the described patients since the target values of 4 and 8 mg/l are proposed for bacterial MIC of 0.062 and 0.125 mg/l, respectively.

Comments on the Quality of English Language

Please check the whole text for spelling mistakes e.g. Abstract line 13 “ clinical settings. of this.”

Author Response

Responses to Reviewer 3

The authors have formulated a well written and important manuscript regarding dalbavancin TDM in clinical practice. An aim is defined in this retrospective analysis, however some points could use some revision:

  1. Cmax-based and trough based: Why do you chose to mix the terminology and not standardize to cmax and cmin oder peak an trough?

We thank the Reviewer for the suggestion. The term “trough” has been replaced by “Cmin” throughout the text.

  1. What software did you use for regression models, statistics and figures? Please state in the method section

The requested information has been added in the revised methods section.

  1. Please explain why you didn’t compare different models for the analysis (Table 1) like it is described in Bhamidipati or why did you chose loglinear since there are different options? Why didn’t you use mean absolute error (MAE)/root mean square error (RMSE) for evaluation?

We can understand the concerns of the Reviewer. After setting up the dalbavancin TDM service in our hospital initially we considered initially different models for the analysis (i.e. linear, loglinear, exponential, etc.). With data accumulation we found that the loglinear function was associated with the best fit and lower error of predictions. For this reason, we have decided to use this function for the estimation of the optimal timing for the next dalbavancin injection in our routine TDM service.   

  1. You describe a retrospective analysis of routine data. So how did you estimate the dosing interval with cmax based TDM within these two years after cmax measurement? What was the standard approach bevor you analysed the log-linear regression models with your data as a result after the 2-year experience? Please describe the routine approach in the method section.

We apologize for the misunderstanding. As better clarified in the revised version of the manuscript, the present study is a retrospective analysis of a 2-year experience with the TDM of dalbavancin. This service, which is routinely carried out in our hospital, includes also the on demand estimation of the time for the next dalbavancin injection. For this estimation we require that physicians collect both the Cmin and the Cmax dalbavancin concentrations. Not all the physicians use this estimation. This is why we described two cohort of patients: one cohort in which TDM was based on both the Cmin and the Cmax measurements (with the estimation of the next dalbavancin dose performed by our service) and one cohort in which TDM was based on Cmin measurement (in this case we do not provide estimations, and physicians decide the time for the next injection).

  1. Please check the formatting for the pathogens e.g. line 43. Shouldn’t that be written in italics?

We apologize for the mistake. Pathogen have been written in italics in the revised manuscript.

  1. The manufacturer recommends a dose adjustment according to GFR. Please add the GFR for the patients in the demographics.

The requested information has been added in the revised manuscript.

  1. What was the standard dosing procedure in your hospital? The actual description is quite confusing. 1000 mg and 500 mg or 1500 mg? 

We apologize for the confusing concepts. As better clarified in the revised manuscript, patients with osteoarticular infections are usually treated with 1500 mg (day 1, day 7, and then the same dose is given every 42-48 days based on TDM results). Patients with GFR < 30 mL/min are treated with 1000 mg (day 1), followed by maintenance doses of 500 mg.

  1. Which pathogens where found in the described patients since the target values of 4 and 8 mg/l are proposed for bacterial MIC of 0.062 and 0.125 mg/l, respectively.

More information on the pathogens identified in the patients with osteoarticular infections have been added in the revised manuscript.

Round 2

Reviewer 1 Report

Comments and Suggestions for Authors

It seems that the authors have successfully  addressed the comments. 

Author Response

It seems that the authors have successfully addressed the comments.

We thank the Reviewer for the positive comment.

Reviewer 3 Report

Comments and Suggestions for Authors

Thank you very much for the revisions. I think they really improved the paper.

Two further comments:
Please also add the software in the statistic section (2.4).

Please check the formula in Table 1. Is there an "x" missing in the "model" column?

Author Response

Thank you very much for the revisions. I think they really improved the paper.

Two further comments:

Please also add the software in the statistic section (2.4).

The software has been added in the statistic section

Please check the formula in Table 1. Is there an "x" missing in the "model" column?

Table 1 has been revised as requested.